Autoregressive models for session-based recommendations using set expansion

Yu Tianhao 1
Zhou Xianghong 2 xhz7@tongji.edu.cn
Deng Xinrong 3
1 University of Shanghai for Science and Technology , Shanghai , China
2 School of Economics and Management, Tongji University , Shanghai , China
3 Fujian BTNG , Fuzhou , China
Mikler Armin
Electronic publication date: 2025 Feb 21
Publication date: 2025
Volume: 11
Electronic Location ID: e2734
Received 2024 Nov 18; Accepted 2025 Feb 5
Copyright: © 2025 Yu et al.
Copyright year: 2025
Copyright holder: Yu et al.
License: This is an open access article distributed under the terms of the Creative Commons Attribution License, which permits unrestricted use, distribution, reproduction and adaptation in any medium and for any purpose provided that it is properly attributed. For attribution, the original author(s), title, publication source (PeerJ Computer Science) and either DOI or URL of the article must be cited.
License URL: https://creativecommons.org/licenses/by/4.0/

Keywords: Set learning, Recommendation system, Autoregressive, Session-based recommendation

Funding: Project of Tongji University Intelligent Recommendation System for Personalized Experience on Interactive Platform: TJW2021553981 This research is supported by Project of Tongji University: Intelligent Recommendation System for Personalized Experience on Interactive Platform (Project No. TJW2021553981). The funders had no role in study design, data collection and analysis, decision to publish, or preparation of the manuscript.

==============================
With the rapid growth of internet technologies, session-based recommendation systems have emerged as a key paradigm in delivering personalized recommendations by capturing users’ dynamic and short-term preferences. Traditional methods predominantly rely on modeling the sequential order of user interactions, deep learning approaches like recurrent neural networks and Transformer architectures. However, these sequence-based models often struggle in scenarios where the order of interactions is ambiguous or unreliable, limiting their real-world applicability. To address this challenge, we propose a novel session-based recommendation model, Deep Set Session-based Recommendation (DSETRec), which approaches the problem from a set-based perspective, eliminating dependence on the interaction sequence. By conceptualizing session data as unordered sets, our model captures the coupling relationships and co-occurrence patterns between items, enhancing prediction accuracy in settings where sequential information is either unavailable or noisy. The model is implemented using a deep autoregressive framework that iteratively masks known elements within a session, predicting and reconstructing additional items based on set data characteristics. Extensive experiments on benchmark datasets show that DSETRec achieves outperforms state-of-the-art baselines. DSETRec achieves a 13.2% and 11.85% improvement in P@20 and MRR@20, respectively, over its sequence-based variant on Yoochoose. Additionally, DSETRec generalizes effectively across both further short and long sessions. These results highlight the robustness of the set-based approach in capturing unordered interaction patterns and adapting to diverse session lengths. This finding provides a foundation for developing more flexible and generalized session-based recommendation systems.

Introduction

With the widespread adoption of internet technology, recommendation systems have become increasingly pivotal across various domains (Gao et al., 2023). As one of the emerging paradigms, session-based recommendation systems have attracted considerable attention in recent years. Unlike traditional recommendation systems, session-based recommendation methods focus on capturing users’ short-term and dynamic preferences to deliver more timely and precise recommendations. This approach has been extensively utilized in the domain of personalized recommendations (Wang et al., 2021).

Early session-based recommendation methods primarily concentrated on item-to-item relationships, such as those utilizing Markov chains (Garcin, Dimitrakakis & Faltings, 2013) and k-nearest neighbors (KNN) (Hariri, Mobasher & Burke, 2012). However, these methods have limitations in capturing the intricate dependencies within the current session, which restricts their recommendation effectiveness. In recent years, deep learning technologies have demonstrated considerable potential. Researchers have proposed session-based recommendation methods leveraging recurrent neural networks, which effectively integrate historical information to enhance recommendation outcomes (Hidasi et al., 2015; Beutel et al., 2018). Additionally, the integration of convolutional neural networks with graph neural networks has further improved the capability to capture complex interactions within sessions (Wu et al., 2019; Liu, Xia & Huang, 2024; Meng et al., 2024). Recently, the introduction of Transformer (Vaswani, 2017) architectures has effectively addressed the limitations in handling long sequence sessions, leading to a notable enhancement in recommendation performance (de Souza Pereira Moreira et al., 2021).

Despite the substantial advancements achieved by deep learning methods in session-based recommendation, a prevalent issue remains. Specifically, these methods generally model historical interactions as sequences, relying on neural networks for feature extraction and user preference prediction (Wang et al., 2021). However, this over-reliance on interaction sequences can introduce biases, potentially leading to misleading recommendations in scenarios where user behaviors deviate from a clear sequential pattern. Such limitations are particularly evident in domains like e-commerce or video streaming, where users often exhibit complex, non-linear, or unordered interaction behaviors. To address these challenges, we propose a set-based recommendation model that conceptualizes historical interactions as an unordered set, enabling the effective modeling of non-sequential user preferences. This approach not only mitigates the biases introduced by sequence dependency but also enhances computational efficiency, making it well-suited for large-scale and heterogeneous datasets.

To handle the unordered nature of the data, we conceptualize observable historical interactions as a set and develop a session-based recommendation model using deep learning. To handle the unordered nature of the data, we conceptualize observable historical interactions as a set and develop a session-based recommendation model using deep learning.

The primary contributions of this article are as follows: We employ a set-based extension method to predict the next item in a session. By viewing sessions as set inputs and capturing the coupling relationships and overall co-occurrence features among set data, it predicts additional elements within the set without relying on the sequence of interactions.

We achieve multi-stage interaction prediction at the session level through an autoregressive approach. The model sequentially masks known elements of the session and then predicts and reconstructs them iteratively using the set-based extension technique.

We compare the proposed model with other state-of-the-art methods, demonstrating the effectiveness of the set-based extension approach in session-based recommendation using various evaluation metrics.

Related work

In this section, we will introduce the concepts of session-based recommendation and set learning related to our research on autoregressive session-based recommendation models with set expansion.

Session-based recommendation

Session-based recommendation is a crucial technology in personalized recommendation, designed to predict user preferences based on implicit feedback within a session, thereby improving user experience and addressing personalized needs. Traditional session-based recommendation methods, such as K-nearest neighbors (Garcin, Dimitrakakis & Faltings, 2013) and Markov chain (Hariri, Mobasher & Burke, 2012) approaches, typically lack sensitivity to the temporal or sequential aspects of interactions. These methods primarily focus on using historical user behavior data to predict items of potential interest. However, when rating data is missing or when the contextual information within sequences cannot be effectively modeled, these methods frequently fall short. This can result in limited applicability in real-world scenarios (Hou et al., 2022).

As research in neural networks progresses, an increasing array of neural network models is being utilized in session-based recommendation. In these deep learning methods, the sequence of interactions is deemed critical. Hidasi et al. (2015) introduced GRU4Rec, a session-based recommendation model leveraging multi-layer gated recurrent units (GRUs). This model effectively captures the sequential dependencies within the data and integrates historical information into the recommendation process, offering valuable insights for future research. Nevertheless, RNN-based methods exhibit limited learning capacity when dealing with long sessions. Recently, graph neural networks (GNNs) have brought new developments to the session-based recommendation field by incorporating the transition relationships between items. Wu et al. (2019) introduced SR-GNN, a model that maps session information into a graph structure and uses GNNs to capture item transition information. Liu et al. (2018) developed STAMP, a short-term memory prioritized recommendation model based on attention mechanisms that integrates general and current interests for joint recommendations. While these methods effectively model complex intra-session dependencies using deep neural networks, they impose stringent requirements on interaction sequences. Currently, research on session-based recommendation methods that do not depend on interaction sequences remains in its nascent stages (Wang et al., 2021). This article builds on these methods by adopting an approximator-aggregator architecture designed for set data, enabling session-based recommendation tasks without relying on interaction sequences.

Set learning

In this research, observable historical interactions within a session are conceptualized as a set, and set learning methods are employed to integrate feature information and user preferences. We model the session-based recommendation task as an expansion of the current set elements. Set learning focuses on predicting the attributes or labels of the output set based on the features of the input set (Wagstaff et al., 2022). In contrast to traditional regression learning methods, set learning encounters challenges like variable set length and unordered elements. Consequently, capturing the dependencies or interactions among elements within a set effectively remains a crucial issue. Given that the unordered nature of sets aligns with our focus on interaction sequences in session-based recommendation, selecting an appropriate set learning method is crucial for extracting session interaction features.

Researchers are currently concentrating on applying set learning within neural networks. For instance, PointNet introduced by Qi et al. (2017) and DeepSets by Zaheer et al. (2017) both convert set vectors into new representations through various pooling techniques. Both methods emphasize modeling set functions. Specifically, PointNet was the first to introduce the concept of unordered sets and employs max pooling to aggregate vector information (Hidasi & Karatzoglou, 2018). Conversely, DeepSets aggregates information by summing vector representations, representing an efficient implementation of the Janossy pooling paradigm (Murphy et al., 2018). Building on this, Skianis et al. (2020) introduced RepSet, which utilizes a new structure to learn set representations, targeting the computation of correspondences between input sets and hidden sets in network flow problems. Furthermore, Lee et al. (2019) introduced the Set Transformer, which employs a Transformer architecture to model interactions between elements in input sets and applies attention mechanisms to set learning, greatly influencing our research. In this study, we apply suitable set learning techniques to expand the elements of the current set, creatively transforming the session-based recommendation problem into a set learning problem for effective resolution.

Model

In this section, we first propose and explain the hypothesis regarding the interaction sequence in session-based recommendations, and then introduce the session-based recommendation model designed to validate the aforementioned hypothesis, detailing the processing steps of each layer of the model.

Problem formulation

In session-based recommendation, we use V={v1,v2,…,v|v|} to represent the total set of items composed of all available items. A session is a sequence of items consisting of interaction behaviors (such as user clicks, views, favorites, etc.), which can be represented as l=[v1,v2,…,vn], where vi∈V(1≤i≤n) are items sorted by timestamp. The next item recommendation in the session can then be defined as v^m+1=grec(l), where grec is the recommendation algorithm.

According to the above description, the goal of session-based recommendation is to identify from the total item set V the next item v^m+1 most likely to be interacted with in the current session si. Previous research has suggested that the realization of the recommended item v^m+1 depends on the items vi sorted by timestamps within the session. However, in real-world scenarios, the dependency of the recommended item v^n+1 on historical items may not necessarily be based on their intrinsic order; that is, the chronological order of items is not a prerequisite. For example, shopping sessions are sometimes unordered, where users may select a series of items without following a clear logical sequence (Wang, Hu & Cao, 2017) (e.g., phone, computer, camera). Historical interactions such as clicking “phone” before “computer” or “computer” before “phone” do not significantly impact the next recommended item, such as “camera”. In such cases, excessive emphasis on the order of interactions may lead to ineffective or incorrect recommendations. To avoid the erroneous impact of sequence on recommendation results, we propose the hypothesis that interactions in session-based recommendations are unordered features.

To validate the above hypothesis, we consider representing session items in the form of a set, thereby satisfying the requirement of unordered features. By representing session items as an interaction set si={v1,v2,…,vm}i, the next item recommendation in the session can be defined as:

(1) v^m+1=frec(s1:m).

Here, s1:m represents the set of items from the first to the m th item in the session, and frec is the set-based learning algorithm.

In set session-based recommendation, it is generally believed that items within the same session share similar attributes (Qi et al., 2017). For example, in the scenario depicted in Fig. 1, if the session includes the known historical items [v1,v2,v3], it indicates that the session’s focus for the current user is on the feature <ω> . <ω> represents the co-occurrence feature of the session mentioned above, and therefore, the recommended item v^4 should also depend on the feature <ω> presented by the session, which is referred to as co-occurrence dependency (Gwadabe & Liu, 2022). Compared to sequential dependency, co-occurrence-based dependency typically exhibits weaker and more ambiguous characteristics. Hence, the objective of our set extension is to fully aggregate the co-occurrence information within the input session items to capture the session’s co-occurrence dependency.

Figure 1 Comparison of session-baesd recommendation between sequential data and ensemble data.

Model overview

In this section, we propose a general session-based recommendation method based on unordered feature networks, called Deep Set Session-based Recommendation (DSETRec), aimed at validating the hypothesis presented above. This model represents sessions in the form of sets and utilizes an approximator-aggregator architecture suitable for set data to extend the current set. The extended elements are used as the recommended items for the session, thereby achieving the prediction of subsequent items in the current session.

As illustrated in Fig. 2, the DSETRec model introduced in this study primarily consists of the following components: input, embedding, set extension, prediction, and output. The set extension component is constructed using an approximator-aggregator structure based on neural networks, while the prediction layer is composed of linear layers. We employed the architectural design proposed in DeepSets (Zaheer et al., 2017) for the set extension, as it aligns with our need to predict set-based data. This design allows the model to simultaneously focus on the historical item information within the session context, forming a globally sensitive receptive field that captures preference changes from a session-wide perspective.

Figure 2 Overview of DSETRec model structure.

During the recommendation process, the DSETRec model converts the session’s contextual information, presented as a set, into low-dimensional embedding vectors via the embedding layer. The set learning layer can utilize any neural network capable of handling sets to learn the co-occurrence information across different subspaces of the embedding vectors, subsequently aggregating them into the session’s overall co-occurrence feature. The prediction layer then outputs the prediction probability Y^ for the total item set V based on these co-occurrence features.

The fundamental concept of the DSETRec model is to capture the co-occurrence features of the current session through set extension without considering the sequential order of the session’s interactions. Based on these co-occurrence features, the model generates “complementary elements” as item preference predictions for the ongoing session. It is important to note that we implemented the set extension method by stacking neural networks with aggregation functions—where the neural network functions as the approximator and the aggregation function serves as the aggregator. The contents of the approximator and aggregator can be freely replaced with any model designed for set-based data. Additionally, the approximator and aggregator differ in that the approximator modifies individual elements within the set, whereas the aggregator combines the elements transformed by the approximator to alter the set as a whole.

The training process of the proposed model also deviates from conventional methods. Specifically, we designed an autoregressive training strategy that progressively masks known elements, expanding and predicting the item set step by step. Overall, DSETRec emphasizes extracting the session’s “co-occurrence features” through set learning and then supplementing the set with elements that best match these “co-occurrence features.” In the subsequent sections of this chapter, we will provide a detailed explanation of the processing mechanisms within each layer of the model (Section, “DSETRec Model”).

DSETRec model

Input and embedding layer

In the model’s input process, to prevent the interaction order from affecting the prediction results, we use a set representation when processing session context information. Specifically, we construct the model’s input as an interaction set s={v1,v2,⋯,vm}, where the elements vi in the set correspond to the interaction items in the original session. By using a set representation, the model’s permutation invariance is ensured, thereby guaranteeing that the output of the embedding layer is not influenced by the order of the input sequence (Wagstaff et al., 2022).

When the input interaction set s passes through the model’s embedding layer, the context information received by the embedding layer consists only of discrete item IDs, which is insufficient for set learning and prediction tasks. To capture the complex relationships between items, we employ a learnable embedding matrix W∈Rd×|I|, where d is a learnable hyperparameter dimension, and |I| is the size of the vocabulary. Each row of the embedding matrix W corresponds to an item, and this matrix maps the ID information into a low-dimensional, dense continuous vector space. Through this embedding process, we obtain the embedded interaction set X∈Rm×d, where the embedding layer can be represented as follows:

(2) xi={W:,i}.

Here, s1:m represents the set of items from the first to the mth item in the session, and frec is the set-based learning algorithm.

Permutation invariance modeling

In traditional session-based recommendation models, recurrent neural networks (RNNs) or graph neural networks (GNNs) are often used to model interaction sequences. However, these methods typically rely on the chronological order of interactions, which can lead to suboptimal recommendation results in certain cases. To overcome this limitation, this study adopts the permutation-invariant design concept from the Deep Sets model (Zaheer et al., 2017), treating the interaction sequence as a set and modeling it through set learning.

Specifically, suppose the input session is a set s={v1,v2,…,vn}, where each element vi is an item in the interaction sequence. To ensure the model’s permutation invariance—that is, the output remains unchanged when the order of the input sequence is altered—the Deep Sets model applies the same mapping operation to each element in the set and then combines these mapped results through an aggregation function. This process can be expressed as:

(3) ϕ(s)=ρ(∑i=1nϕ(vi))

where the mapping function ϕ is a universal approximator, and the aggregation function ρ aggregates the mapping results of all elements.

According to the Deep Sets theory (Zaheer et al., 2017), as long as the aggregation function is symmetric (such as summation, averaging, etc.), the model can remain invariant to the input order. This design allows the model to focus on capturing the co-occurrence features within the set without relying on the temporal order of the items.

Set extension

As described in “Model Overview”, the set extension module focuses on extracting co-occurrence information from the session embedding vectors X to capture the overall co-occurrence features of the session (Zhao et al., 2023). The embedding vectors X form a set of d-dimensional elements. Based on the model design in “DSETRec Model”, the model learns the entire set during the set learning layer while maintaining permutation invariance (Lee et al., 2019). Permutation invariance refers to the property that, if there is a function f that operates on the set X, the result of the function is unaffected by the order of elements in the set (Zaheer et al., 2017).

This means that in our model’s set learning layer, the learning outcome remains unaffected by the order in which the items are processed, which can be expressed as:

(p1,p2,⋯,pn)≠(q1,q2,⋯,qn)SL({vp1,vp2,⋯,vpn})=SL({vq1,vq2,⋯,vqn})

where (p1,p2,⋯,pn) and (q1,q2,⋯,qn) represent two different permutations, and SL is the set learning algorithm.

This characteristic ensures that when the input embedding vectors X are presented, the set learning layer can capture the same features regardless of the order of the data. According to research (Zaheer et al., 2017), if a function f:X→Y has permutation invariance, it is necessary and sufficient that the function can be decomposed as ρ(∑x∈Xφ(x)), where ρ and φ can represent arbitrary functions or universal approximators.

The overall structure of the set learning is shown in Fig. 3. The core of set extension lies in effectively extending the existing set to predict the missing elements in the set. For session-based recommendation tasks, this means predicting the next possible interaction item vn+1 based on the historical interaction item set s={v1,v2,⋯,vn}. The set extension strategy based on Deep Sets can be implemented as follows.

Figure 3 Set extension based on permutation invariance modeling.

First, we map and aggregate the input set s: Considering that each stage can be viewed as an independent set and that the order of elements in these sets should not affect the overall prediction results, we incorporate the concept of permutation invariance into the model and model it. Let X1,X2,…,Xn represent the input sets at each stage, and the model must ensure that when any permutation π is applied to the input sets, the output remains unchanged. Specifically, we can define the permutation-invariant function as:

(4) f(X1,X2,…,Xn)=ρ(∑i=1nϕ(Xi))

where ϕ is a feature extraction function for each set element, and ρ is a nonlinear transformation used to aggregate the multi-stage results. Through this architecture, we can effectively achieve permutation-invariant modeling of multi-stage inputs, ensuring the model’s stability when faced with inputs in different orders.

When processing co-occurrence information between set elements, we successively use two deep neural networks as universal approximators to complete the set learning task. To ensure that this set-based method can be widely applied to various session-based recommendation tasks, we used a relatively simple fully connected neural network as our approximator, although any deep neural network can be substituted. During the set learning layer processing, the input vectors X are first mapped to a new space by a universal approximator φ for each element in the set, and then these mapped elements are aggregated through an aggregation operation. The resulting output is further mapped through a second approximator ρ for the next step.

In practice, the universal approximator φ chosen in this study is a stack of two fully connected neural networks. During the processing of these neural networks, we also introduced two residual connections (He et al., 2016) to further enhance the extraction of co-occurrence features from the elements in the set by the universal approximator and to prevent potential issues of vanishing or exploding gradients. The choice of different aggregation methods following the approximator φ can also affect the model’s convergence speed, which we further discuss in the experimental section.

In the universal approximator φ, we also use an aggregation operation, denoted as ϱ. We can express the universal approximator φ as follows:

(5) DNNi=WiF+biφ=DNN2(ϱ(DNN1(X)+X))+ϱ(DNN1(X)+X))

where DNNi represents the ith deep neural network in the universal approximator φ, and Wi and bi represent the corresponding parameter matrix and bias vector, respectively. ϱ is the aggregation operation.

After passing the input set {x1,x2,…,xn} obtained from the embedding layer through the universal approximator φ, it can be transformed into φ(x1),φ(x2),…,φ(xn). We then continue by passing these through the aggregator ρ, which includes the aggregation operation ς and a second neural network.

(6) ρ=WjF+bjF=ρς(φ(x1),φ(x2),⋯,φ(xn))

where ς represents the aggregation operation, including summation or averaging aggregation operations. ρ is the aggregator composed of a fully connected network, and Wj and bj are its corresponding parameter matrix and bias vector.

The aggregator differs from the approximator in that it implements changes to the entire set that has been mapped by the approximator. It is worth noting that the aggregator also needs to satisfy the “symmetric design,” meaning it should avoid the influence of the input order of elements on the set. Therefore, we use summation or averaging as the aggregation methods here, and we further discuss the performance impact of different aggregation methods in subsequent experiments. After passing through the aggregator ρ, we obtain the aggregation vector F and consider it to contain the co-occurrence features of the session <D>, which we continue to use in the prediction layer (Wu et al., 2020).

Finally, we use the prediction head to transform the aggregated co-occurrence information into probability output. The prediction head is essentially a linear layer, and this mapping operation consists of a weight matrix and an optional bias term. After the interaction items in the session s pass through the set learning layer to obtain the final output F, we consider that F contains the co-occurrence features learned from the current session. The prediction head maps the output F, containing the co-occurrence features, to a dimension corresponding to the total item set V and ultimately outputs the probabilities through the softmax function.

(7) Y^=softmax(WpF+bp)

where Wp and bp are the parameter matrix and bias vector of the prediction head. We share the weights between the embedding layer and the prediction head to alleviate overfitting and reduce model size.

At this point, our model successfully maps the co-occurrence features in the interaction set s into recommendation probabilities in the total item set V and outputs the probability list Y^=y^1,y^2,…,y^|v|, which contains the predicted scores for all possible interaction items in the total item set V. From the probability list Y^, we select the interaction item vi corresponding to the highest predicted score y^i, which is the next recommended item vm+1 for the session.

Multi-stage autoregressive prediction

In the DSETRec model, autoregressive prediction is achieved by gradually extending the current session set. Specifically, we first apply a masking operation to the current session s={v1,v2,…,vn} to generate a partially observable set S′. Then, using the set extension method, we predict the missing items in the set and progressively add them to the current set.

Let S(t) represent the state of the session set at step t, then the autoregressive prediction can be expressed as:

(8) S(t+1)=S(t)∪v(t+1)

where v(t+1) is the next item predicted by the set extension method: v(t+1)=arg⁡maxv∈VP(v|S(t)). This process iterates until a complete recommendation sequence is generated.

During the prediction phase, for each stage’s input, we introduce a masking operation that progressively hides parts of the input, forming an autoregressive chain structure. Specifically, let the input set at the current stage be Xm; the model applies a masking operation M to generate partially observed data:

(9) Xm=M(Xm).

Then, the set extension mechanism is used to predict the remaining parts of the set:

(10) Xm=ψ(Xm)

where ψ represents the set extension prediction function. This process relies on the results of the previous stage and iteratively updates according to the following autoregressive formula:

(11) Xm+1=g(Xm,Xm−1).

At each stage, by stacking nonlinear activation functions and attention mechanisms, the model enhances its ability to capture long-term dependencies. In particular, the attention mechanism can weigh and sum the input through the weight matrix Am:

(12) Zm=Am⋅Xm.

Through the above steps, this method not only effectively reduces information redundancy in multi-stage prediction but also improves the model’s stability and accuracy when handling long sequences.

Additionally, the loss function used during the training of the model is Top 1 loss (Hidasi et al., 2015). This loss function allows the model to select only the detection result with the highest confidence as the positive sample, while other detection results are treated as negative samples. Previous studies and our experiments have demonstrated the effectiveness of this loss function (Hidasi & Karatzoglou, 2018). The loss function is as follows:

(13) Ltop1=1Ns∑j=1Nsσ(r^s,j−r^s,i)+σ(r^s,j2).

Experiments

In this section, we validate the proposed hypothesis concerning interaction order through comparative experiments. Furthermore, we compare our proposed model with advanced baselines, analyze the strengths and weaknesses of our model, and investigate the impact of hyperparameters on its performance.

Datasets and data preprocessing

To validate the hypothesis proposed in this article and evaluate the effectiveness of the general session-based recommendation model DSETRec, we conducted a series of experiments using the publicly available datasets Yoochoose and Diginetica. The Yoochoose dataset originates from the RecSys 2015 Data Mining Challenge and contains click and purchase events from e-commerce user sessions. Additionally, we selected data closer to the test set for training, rather than training on the entire dataset, to achieve better experimental results (Wu et al., 2019). The Diginetica dataset was sourced from the 2016 CIKM Cup, and we only used its transaction data. Table 1 lists the key information about the relevant datasets.

Table 1 Experimental dataset statistics.

Dataset	Yoochoose (1/64)	Yoochoose (1/4)	Diginetica	
Click	557,248	8,326,407	982,961	
Item	16,766	29,618	43,097	
Train	369,859	5,917,745	719,470	
Test	55,898	55,898	600,858	

Baselines

To validate the performance of the unordered session-based recommendation model proposed in this article, the following models were used as baselines in the experiments. The proposed model was compared with these baselines on the two datasets mentioned above. The baselines are grouped into two categories: traditional recommendation methods (1–4) and sequence neural network-based recommendation methods (5–7): (1) POP: Provides the simplest session-based recommendation by suggesting the most frequently interacted items in the training set.

(2) S-POP: Recommends the most frequently occurring interaction items within the current session.

(3) Item-KNN: Recommends items similar to those previously clicked in the session, with similarity defined by the cosine similarity between session vectors.

(4) FPMC: A sequence prediction model that combines matrix factorization and Markov chains for hybrid recommendation of the next item.

(5) GRU4REC (Hidasi et al., 2015): Models user sequences in session-based recommendation using RNNs and employs parallel mini-batch training with a ranking loss function.

(6) STAMP (Liu et al., 2018): Combines long-term and short-term interests of the session to generate the final session representation.

(7) SR-GNN (Wu et al., 2019): Models item sequences within sessions using Graph Neural Networks (GNNs) and leverages attention mechanisms to obtain session representations.

Evaluation metrics

To facilitate performance comparison with the baseline models, we use the commonly adopted metrics of precision (P) and mean reciprocal rank (MRR) as evaluation criteria. Since in practical recommendation systems, multiple items are typically recommended simultaneously, we use P@K and MRR@K to assess the performance of the models, where K represents the number of recommended items.

P@K: Precision (P) is widely used as a measure of prediction accuracy in the session-based recommendation field. P@K represents the proportion of test cases with correct recommended items in the top K positions of the ranked list. In this article, P@20 is mainly used, defined as:

(14) P@K=nhitN

where N denotes the number of test cases in the session-based recommendation task, and nhit is the number of cases where the desired item appears in the top K of the ranked list. A prediction is considered accurate if the recommended item appears in the top K positions.

MRR@K: Mean reciprocal rank (MRR) is the average of the reciprocal ranks of the correct recommended items; if the rank exceeds K, the reciprocal rank is set to 0. In some experiments in this article, MRR@20 is used, defined as:

(15) MRR@K=1N∑vtarget∈V¨1Rank(vtarget)

where Rank(vtarget) is the rank of the target item vtarget in the total item set V¨.

Both P and MRR are normalized scores in the range of [0, 1], considering the order of recommendation rankings. Higher values indicate that the correct recommendations are ranked higher in the list, signifying better performance of the corresponding recommendation system.

Hyperparameter settings

The hyperparameters of the proposed session-based recommendation model DSETRec were optimized and determined through grid search and early stopping. The final hyperparameters are as follows: model dimension d=256, training batch size batchsize=64, and the Adam optimizer was used to optimize these parameters during training with mini-batches. Additionally, due to the large size of the datasets and the high number of iterations, the initial learning rate lr was set to 10−4 and was reduced to 5×10−5 and 10−5 after several iterations.

Experimental results

Comparison of performance between set data and sequential data

To further validate our proposed hypothesis, this section compares the performance of the DSETRec model with position encoding, representing “ordered” data (Romero & Cordonnier, 2020), against the DSETRec model without position encoding. Comparative experiments were conducted on the two commonly used datasets mentioned earlier, and the results are shown in Table 2. We have highlighted the values where the performance of the ordered and unordered models is relatively better in bold.

Table 2 Comparison of unordered and ordered performance under DSETRec model.

Bold indicates the values where the performance of the ordered and unordered models is relatively better.

Model	DSETRec (set)	DSETRec (sequence)	Gap	
Datasets	Metric (%)	
Yoochoose 1/64	P@10	63.26	50.06	+13.2	
	MRR@10	38.92	27.07	+11.85	
	P@20	70.70	58.89	+11.81	
	MRR@20	39.43	28.15	+11.28	
Yoochoose 1/4	P@10	62.88	51.66	+11.22	
	MRR@10	36.84	27.38	+9.46	
	P@20	69.84	60.05	+9.79	
	MRR@20	37.33	28.23	+9.1	
Diginetica	P@10	39.88	38.02	+1.86	
	MRR@10	21.41	18.99	+2.42	
	P@20	48.87	44.19	+4.68	
	MRR@20	21.83	16.89	+4.94	

As shown in the table above, when using the DSETRec model, the unordered set model without position encoding consistently outperforms the ordered sequence model with position encoding across all datasets. This indicates that our DSETRec model is more adept at handling set-type data.

For the session-based recommendation tasks on the above datasets, the DSETRec model may not be sensitive to the positional information of the interaction items. In practical recommendations, the order of item interactions within the same session can be arbitrary and incidental, with the interaction order not necessarily guiding the recommendation results. Therefore, introducing position encoding does not provide the model with critical information and may even introduce noise, reducing the model’s performance. The presence of position encoding, representing “order,” hinders the aggregation of co-occurrence information among session items. This “hindrance” effect is particularly evident in the more easily predictable Yoochoose dataset (Chen, Li & Zhou, 2021).

Model performance analysis

To further demonstrate the performance of the set-based learning recommendation algorithm proposed in this article, we compared the DSETRec model with other recommendation algorithms or models. To ensure a fair comparison, the Baseline methods retained the original data preprocessing, filtering out items with fewer than five interactions. The results are shown in Table 3. In order to better observe the performance comparison with the baselines, we have highlighted the results of the DSETRec model in bold and underlined the best-performing results.

Table 3 Comparison of model performance with baselines.

In order to better observe the performance comparison with the baselines, we have highlighted the results of the DSETRec model in bold and underlined the best-performing results.

Model	Yoochoose 1/64	Yoochoose 1/4	Diginetica	
	P@20 (%)	MRR@20 (%)	P@20 (%)	MRR@20 (%)	P@20 (%)	MRR@20 (%)	
POP	6.71	1.65	1.33	0.30	0.89	0.20	
S-POP	30.44	18.35	27.08	17.75	21.06	13.68	
FPMC	45.62	15.01	–	–	26.53	6.95	
Item-KNN	51.60	21.81	52.31	21.70	35.75	11.57	
GRU4REC	60.64	22.89	59.53	22.60	29.45	8.33	
STAMP	68.74	29.67	70.44	30.00	45.64	14.32	
SR-GNN	70.57	30.94	71.36	31.89	50.57	17.59	
DSETRec	70.70	39.43	69.84	37.33	48.87	21.83	

When comparing with other algorithms or models, it is clear that the DSETRec model proposed in this article outperforms all the classical recommendation algorithms and most deep learning models mentioned, providing strong validation for the effectiveness of our proposed recommendation algorithm. However, despite this, our model shows some gaps in Precision compared to some state-of-the-art models like SR-GNN. Nevertheless, in terms of the MRR metric, our model performs exceptionally well, surpassing all other leading models in the field. This could be attributed to our model’s training approach, which applies each item more frequently across the dataset than the training methods of other models, playing a crucial role in capturing the co-occurrence features of the session.

Overall, the DSETRec model, based on a general architecture, demonstrates strong performance and provides preliminary validation for the feasibility of set-based learning in session-based recommendation. This also underscores the importance of session-wide co-occurrence features, which contribute more significantly to recommendation effectiveness compared to the sequential features among items.

Long and short session performance analysis

To validate the generalizability of our model across sessions of different lengths, we conducted performance comparison experiments on the DSETRec model for sessions of varying lengths. We used the total dataset, which includes all sessions, as the performance benchmark. Sessions with five or fewer items were defined as short sessions, while those with more than five items were defined as long sessions. To further evaluate the model’s adaptability to different session lengths, we chose K=20 as a key configuration. This choice is based on the fact that extremely short sessions (e.g., fewer than five items) may lack sufficient information, resulting in reduced model performance when understanding user behavior and making accurate recommendations. On the other hand, excessively long sessions may contain redundant information, increasing the model’s computational complexity and potentially overwhelming it with unnecessary details. By selecting K=20, we aimed to strike a balance between leveraging enough contextual information and avoiding performance bottlenecks from overly short or long sessions. With K=20, we applied the DSETRec model separately to the total sessions, long sessions, and short sessions, and compared the gap in Precision between the long and short session datasets. The performance of the model on sessions of different lengths is shown in Table 4.

Table 4 The gap between different models on long and short datasets.

Model	Yoochoose 1/64	Diginetica	
	All	Short	Long	All	Short	Long	
NARM	68.32	71.44	60.79	62.58	51.22	45.75	
STAMP	68.74	70.69	64.73	45.64	47.26	40.39	
DSETRec	70.70	72.49	69.08	45.87	54.27	43.37	
Gap			1.79 (2.47%)			2.5 (5.45%)	

As seen in the table above, the performance of the DSETRec model on the long session dataset only decreased by about 5% compared to its performance on the overall dataset, indicating that it still maintains good effectiveness in handling long sessions. Comparisons with other models (Liu et al., 2018; Li et al., 2017) show that our model exhibits the highest stability, further confirming our hypothesis for session-based recommendation.

Hyperparameter sensitivity analysis

In this section, we analyze the impact of hyperparameters on the performance of the DSETRec model. To more intuitively observe the model’s performance under different hyperparameter settings, we selected an unfiltered dataset for this experiment. By adjusting the model’s dimension and batch size, we conducted corresponding experiments to explore their effects on precision (P) and mean reciprocal rank (MRR). With K=20, the results of the analysis experiments are shown in the Fig. 4.

Figure 4 The impact of dimensions on metrics.

Regarding the dimensionality, our model tends to prefer higher dimensions. Higher dimensionality allows the model to learn co-occurrence information from different subspaces, which promotes the aggregation of co-occurrence features. It is worth noting that the improvement in model performance is not proportional to the increase in dimensionality. Our model reaches near saturation at 256 dimensions, with further increases in dimensionality providing very limited gains. For the Diginetica dataset, this saturation effect is even more pronounced, with the P and MRR metrics showing different levels of decline at 512 dimensions. As shown in Fig. 5, compared to model dimensionality, batch size has a relatively smaller impact on the model’s performance, showing an overall trend of initial decline followed by a slow rise. Our model tends to perform better with smaller batch sizes. A preliminary analysis suggests that this may be because our gradual training approach during data training is somewhat similar to batch processing, leading to a degree of redundancy that affects certain aspects of performance.

Figure 5 The impact of batch size on metrics.

Aggregation method analysis

In our model, we employed two aggregation operations (the aggregation operation ϱ and the aggregation performed by the aggregator ρ). Common aggregation methods include summation, averaging, and taking the maximum or minimum values. The choice of aggregation method in the experiments had a noticeable impact on our model’s performance. The results for different aggregation methods are illustrated in the Fig. 6.

Figure 6 The impact of different aggregation methods on performance.

In the figure, AVG + SUM indicates that averaging aggregation was applied first, followed by summation aggregation, while AVG++ indicates that averaging was used for both aggregation steps, and so on. The figure shows that using averaging first, followed by summation, provides the best performance for our model. Specifically, for the aggregation operation ϱ, averaging helps capture the details within the input vectors X. For the aggregator ρ, which is at the final stage of the model, summation is effective in retaining more co-occurrence information.

Ablation study

To further explore the effectiveness of the autoregressive and set extension modules in our proposed model, we conducted ablation experiments on the Yoochoose and Diginetica datasets. The results are shown in Fig. 7.

Figure 7 The impact of model components on performance.

The ablation study results demonstrate that removing either the set extension module (w/o set extension) or the autoregressive module (w/o autoregressive) leads to a noticeable decline in the model’s performance across both P@20 and MRR@20 metrics. Among the two, the absence of the set extension module has a more significant impact on the results, indicating its critical role in enhancing the recommendation quality.

This suggests that the set extension module is essential for capturing complex user interaction patterns, while the autoregressive module contributes to the sequential dependency modeling. The complete DSETRec model consistently achieves the best performance, underscoring the effectiveness of the designed architecture and the complementary contributions of these two modules.

Conclusion

In this article, we propose DSETRec, a session-based recommendation model tailored for set-type data. By adopting an autoregressive training approach, the model sequentially predicts and recommends complementary elements within a set, effectively mitigating the adverse effects of interaction order noise on recommendation performance. Unlike traditional sequence-based methods, DSETRec leverages a set-based representation to capture co-occurrence relationships among items without relying on interaction order. This innovative approach enhances the model’s robustness and adaptability, particularly in scenarios where user interactions are unordered or exhibit ambiguous patterns.

To validate the effectiveness of DSETRec, we instantiated the model using fully connected neural networks and conducted extensive experiments on multiple datasets. Our results demonstrated that the model successfully learned and aggregated item features within sessions, achieving superior performance compared to advanced baselines in terms of P and MRR. Notably, experiments evaluating the impact of position encoding further highlighted the advantages of DSETRec in handling set-type data (Romero & Cordonnier, 2020). Additionally, through detailed case studies, we showcased the model’s practical applicability, particularly for medium and long session datasets, where unordered interaction patterns are prevalent.

The contributions of this work extend beyond performance improvements. DSETRec offers a novel perspective on modeling user interactions in recommendation systems, addressing critical limitations of sequence-based methods. By providing a framework that effectively captures unordered interaction dynamics, our model has significant potential for deployment in real-world scenarios, such as e-commerce and content streaming platforms, where user behaviors often defy strict sequential order.

Future research could explore extending the set-based paradigm to other recommendation tasks, including multi-modal recommendations and cross-domain applications. Moreover, incorporating additional mechanisms to enhance interpretability and scalability would further strengthen the practical impact of this approach, contributing to the ongoing evolution of recommendation systems.

Supplemental Information

Supplemental Information 1 Decaying of the dataset and code of the paper.

Additional Information and Declarations

Competing Interests

Xinrong Deng was employed by Fujian BTNG. The authors declare that they have no competing interests.

Author Contributions

Tianhao Yu conceived and designed the experiments, performed the experiments, performed the computation work, prepared figures and/or tables, authored or reviewed drafts of the article, and approved the final draft.

Xianghong Zhou performed the experiments, analyzed the data, prepared figures and/or tables, and approved the final draft.

Xinrong Deng conceived and designed the experiments, analyzed the data, performed the computation work, authored or reviewed drafts of the article, and approved the final draft.

Data Availability

The following information was supplied regarding data availability:

The datasets are available at Zenodo: 俞, . 天浩 . (2025). Research on Autoregressive Session-Based Recommendation Models with Set Expansion [Data set]. Zenodo. https://doi.org/10.5281/zenodo.14673298.

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
