# Peer review of "Autoregressive models for session-based recommendations using set expansion"

_PeerJ Computer Science, doi:10.7717/peerj-cs.2734_

## Round 0.1 · original submission · Major Revisions

Your manuscript has been reviewed by three reviewers, who have articulated a number of required changes. Many comments pertain to stylistic issues, but in some cases, the reviewers have pointed out that additional information/clarification is needed. In particular, the inclusion of some recent related work has been suggested. make the requested revisions and provide a separate summary page of how the reviewers' comments have been addressed.

·

Basic reporting

These are the comments to be addressed
1. The title of the manuscript is to be rewritten as Autoregressive Models for Session-Based Recommendations Using Set Expansion.
2. The abstract should include results and a discussion session
3. Implication of the study missing.
4.in page no 21, authors used (With K = 20, we applied the DSETRec
model separately to the total sessions, long sessions, and short sessions,) why K =20 ? justify.

Experimental design

no comment'

Validity of the findings

no comment'

Additional comments

no comment'

Reviewer 2 ·

Basic reporting

Interesting work, well structured. Bibliography is up-to-date.

Experimental design

It seems to me that the following questions should be reflected:

1. The authors' initial premise: "However, these sequence-based
models often struggle in scenarios where the order of interactions is ambiguous or unreliable"
For sequence-based models, the sequence is decisive. The model restores exactly these dependencies. What dependencies does the proposed model restore? If the next element in the session is completely random, then it is impossible to restore it. This means that the proposed model assumes some dependencies. Which ones? The same question in other words: how to build an explanation for the proposed model? It seems to me that the cited work [12 ]contains thoughts on this matter.
Actually, it is on this basic assumption that the proposed approach should be compared with other models. There may be deep dependencies on the structure and set of features of individual elements. For example, what happens if the elements in the recommender system have a very small number of features (e.g. just one - price)? Or the attributes of the elements are not physically comparable (price and linear dimensions), etc.

2. Many models from the baseline do not use sequence information. For example:
(1) POP: Provides the simplest session-based recommendation by suggesting the most frequently interacted items in the training set.
(2) S-POP: Recommends the most frequently occurring interaction items within the current session.

Validity of the findings

See the above-mentioned questions

Cite this review as

Reviewer 3 ·

Basic reporting

The paper titled "Autoregressive Session-Based Recommendation Models Based on Set Expansion" presents a novel approach to session-based recommendation systems, addressing the limitations of traditional sequential models. The authors introduce DSETRec, a model that conceptualizes session data as unordered sets, thereby eliminating the reliance on the sequence of user interactions. employing a deep autoregressive framework, the model captures co-occurrence patterns and coupling relationships between items, which enhances recommendation accuracy in scenarios where sequential information is ambiguous or unreliable. The experimental results demonstrate that DSETRec outperforms existing state-of-the-art methods across multiple benchmark datasets, confirming the potential of the set-based approach in improving session-based recommendation systems. Some improvements are shown below:

1. Pay attention to the standardization of citation subscripts in literature. For example, in Chapter 4.1, it is stated that "Additionally, inspired by the experimental method in [8]." Please review the rest of the text.
2. Add charts and visual elements to help explain complex concepts and showcase experimental results. Figure 1 only shows the substitution of sequential sequences in the form of sets, and does not represent the permutation invariant design concepts of co-occurrence dependencies or deep set models. Visual elements can be added to Figure 1.
3.Although this article compared DSETSec with several baselines, there is a lack of in-depth ablation studies to separate the contributions of various components in the model. Can additional ablation studies be conducted to demonstrate the individual contributions of set extension techniques and autoregressive frameworks to the overall performance of DSETRec.
4. The references can be updated to include some recently published related works
5. The colon in the title of the image can be replaced by a dot.
6. The conclusion section should be more comprehensive, emphasizing the practical application value of the model and its contribution to the field of recommendation systems, making the conclusions and contributions of the paper clearer.
7. Does the DSETSec model have limitations in certain scenarios? If so, please fully discuss the possible underperformance compared to traditional methods.

Experimental design

Please refer to "1. Basic reporting"

Validity of the findings

Please refer to "1. Basic reporting"

Additional comments

Please refer to "1. Basic reporting"

Cite this review as

---

## Round 0.2 · accepted · Accept

I have read the response to reviewers' comments, and I am confident that all the concerns have been addressed. Thank you for clearly indicating the edits that have been made in the resubmitted version of the manuscript. This manuscript is ready for publication.